# Retention Rates of Genetic Therapies Based on AAV Serotypes 2 and 8 Using Different Drug-Delivery Materials

**DOI:** 10.3390/ijms25073705

**Published:** 2024-03-26

**Authors:** Felix F. Reichel, Peter Kiraly, Immanuel P. Seitz, M. Dominik Fischer

**Affiliations:** 1Nuffield Laboratory of Ophthalmology, Department of Clinical Neurosciences, University of Oxford, Oxford OX1 2JD, UK; felix.reichel@med.uni-tuebingen.de (F.F.R.); peter.kiraly@ouh.nhs.uk (P.K.); 2Oxford Eye Hospital, Oxford University Hospitals NHS Foundation Trust, Oxford OX3 9DU, UK; 3Centre for Ophthalmology, University Eye Hospital Tübingen, 72076 Tübingen, Germany; immanuel.seitz@med.uni-tuebingen.de

**Keywords:** retention, AAV, gene therapy, subretinal injection, drug delivery

## Abstract

The purpose of this study was to compare the retention rate of Adeno-associated viral vector (AAV) gene therapy agents within different subretinal injection systems. The retention of AAV serotype 2-based voretigene neparvovec (VN) and a clinical-grade AAV serotype 8 vector within four different subretinal cannulas from two different manufacturers was quantified. A standardized qPCR using the universal inverted terminal repeats as a target sequence was developed. The instruments compared were the PolyTip^®^ cannula 25 g/38 g by MedOne Surgical, Inc., Sarasota, FL, USA, and three different subretinal injection needles by DORC, Zuidland, The Netherlands (1270.EXT Extendible 41G subretinal injection needle (23G), DORC 1270.06 23G Dual bore injection cannula, DORC 27G Subretinal injection cannula). The retention rate of VN and within the DORC products (10–28%) was comparable to the retention rate (32%) found for the PolyTip^®^ cannula that is mentioned in the FDA-approved prescribing information for VN. For the AAV8 vector, the PolyTip^®^ cannula showed a retention rate of 14%, and a similar retention rate of 3–16% was found for the DORC products (test–retest variability: mean 4.5%, range 2.5–20.2%). As all the instruments tested showed comparable retention rates, they seem to be equally compatible with AAV2- and AAV8-based gene therapy agents.

## 1. Introduction

For inherited retinal dystrophies (IRDs), gene therapy is a promising treatment concept that has gained much attention, with voretigene neparvovec (VN) the first ever FDA-approved gene therapeutic agent for IRDs [1]. VN, branded as Luxturna, targets IRDs due to mutations in the *RPE65* gene and is classified as an advanced therapy medicinal product (ATMP). The genetically modified organism (GMO), on the basis of an Adeno-associated virus (AAV), is applied through a subretinal injection. The AAV is used as a vector to deliver a functional copy of the mutated gene into the retinal pigment epithelial (RPE) cells. The success of this treatment has paved the way for a growing number of clinical trials and increased efforts to translate the success of VN to a wider range of other ocular conditions. Today, gene therapy agents are in the clinical trial stage for a broad spectrum of retinal diseases, ranging from chorioretinal dystrophies, like choroideremia, to age-related maculopathies, like age-related macular degeneration (AMD) [2,3,4,5].

Although there are other viral vector platforms like adenovirus or lentivirus to deliver a gene sequence to a target cell, AAVs are the most established viral vectors for gene therapy [6].

Conducive to this success of AAV as a viral vector is the fact that it is non-pathogenic and has low immunogenicity. Furthermore, AAV is replication-deficient in the absence of a helper virus and is non-integrating in the absence of its own genome, which is deleted to make room for the therapeutic payload. This reduces the risk of possible mutagenesis in the host cell. The capsid epitopes of AAVs are decisive for its cell tropism and can be used in gene therapy to selectively target cells, by choosing or designing AAV serotypes with a specific tropism. For retinal gene therapy, the naturally occurring AAV serotype 2 was the first used in clinical trials, and VN is also based on this serotype. AAV8 is another serotype that has been repeatedly used in ocular clinical gene therapy trials, due the strong affinity of this serotype to photoreceptor cells [7].

VN and the majority of retinal gene therapy agents in trials are administered via a subretinal injection. Alternative routes of administration include intravitreal or suprachoroidal injections of the vector. However, as VN and most gene therapy trials target the RPE or photoreceptor cells, the subretinal approach has the advantage of delivering the therapeutic agent into direct contact with the target cells, thus increasing the chance of transduction and limiting off-target effects. Furthermore, the subretinal space is known to be specifically immune-privileged, which reduces the risk of a clinically significant immune reaction to the compound [8,9]. In order to perform a subretinal delivery of a gene therapy agent, a vitrectomy with a posterior vitreous detachment is performed, followed by the injection of the agent via a small-gauge subretinal injection cannula. An injection at the superior arcade was shown to be ideal for the location of the retinotomy and initiation of the bleb, if the macula is the desired area of treatment [10]. Subretinal injections are also performed outside of the gene therapy application setting. They are, for example, established procedures for the subretinal delivery of recombinant tissue plasminogen activator (rTPA) for large submacular hemorrhages in AMD. The materials used for subretinal injections are therefore readily available from a range of different manufacturers. AAVs, however, have specific surface properties that can lead to significant retention rates within these injection devices [11,12,13]. Only two injection needles and two extension tubes are listed in the Surgical Manual for the surgical administration of VN. Procurement of these particular items is not always straightforward for individual health care professionals across the globe. Additionally, it seems likely that other products are equally suitable for the application of VN (and related AAV-based therapies), but simply have not been tested for compatibility. Additionally, surgical technique is developing in this highly innovative space, and the dual surgeon approach, as described for VN, is not always being used. Instead, a foot pedal-controlled injection system is often preferred, leaving health care professionals in doubt as to the biocompatibility of instruments otherwise available for subretinal surgery. Here, compatibility in terms of retention rates was tested in a range of different subretinal injection cannulas. The retention of AAV2-based VN is compared with a clinical-grade ATMP based on an AAV8 vector.

## 2. Results

The retention rates of the AAV Serotype 2-based VN and the AAV Serotype 8-based IND are given in Figure 1. The retention rates of the subretinal injection needles from D.O.R.C. 1270.EXT, 1270.06, and 1270.04 of the AAV2-based viral vector (Luxturna) were 28.37%, 10.24%, and 19.24%. These were comparable to the retention rate (32.28%) found for the PolyTip^®^ cannula 25 g/38 g that is mentioned in the FDA-approved prescribing information for VN. For the AAV8 vector, the PolyTip^®^ cannula showed a retention rate of 14.28%, which was also similar to the retention rate of the 1270.EXT (8.44%), 1270.06 (15.94%), and the 1270.04 (2.61%). While a small difference between the serotypes was found between the retention rates measured as a percentage of the direct (initial) concentration, no significant difference was found in absolute vector genomes lost within the injection systems (Figure 2 and Figure 3).

## 3. Discussion

Within this study, we compared the retention rates of VN and an AAV8-based IND within four different subretinal injection needles. We show that all four microneedles have comparable retention rates and seem, therefore, equally suitable for the application of AAV2- and AAV8-based vectors.

Previous studies have shown significant differences in the recovery rates from AAV-based gene therapeutic products between application devices [11,12,13]. The application of precise and consistent doses of AAV is, however, critical to ensure the efficacy of the treatment, while limiting off-target effects and the risk of an immune response. Several studies have shown that AAV can induce a dose-dependent immune response in the eye [14,15,16].

A method of reducing the loss of vector within the injection system is the addition of a non-ionic surfactant [11,12,13]. The study from Bennicelli et al. was the first to report that the addition of the surfactant PF-68 could prevent up to 75% of vector loss due to surface adherence [11]. Our group found similar results for an rAAV8-based viral vector and showed that, within a 0.001% PF-68-containing vector solution, more than twice as many vector genome copies can be found after passage through the injection device [12]. PF-68 is a non-ionic tri-block copolymer with a central hydrophobic component flanked by two hydrophilic components. Several copolymers aggregate to micelles, which are believed to shield the AAV epitopes from interaction with the cannula walls. The surfactant was also included in the clinical trials of VN. That being the case, VN was FDA-approved with a supplementary diluent containing 0.001% poloxamer 188, a non-proprietary analog of PF68. And the AAV8-based IND used in this experiment is also supplemented with a diluent containing PF68.

The retention rate and the effect of the PF68 surfactant in earlier studies was shown to be less pronounced for the AAV8 serotype [12]. Here, we have also found slight differences between the AAV2 and the AAV8 serotype. While the retention of VN was, on average (all injections systems combined), 22.5% (±9.4), the average retention of the AAV8-based IND was only 10.3% (±8.6) (*p* = 0.017, Figure 2). The absolute number of mean vector genome copies lost within all injection systems was not significantly different between the two vectors (Figure 3). An explanation for this could be the higher concentration of the AAV8-based IND (1 × 10^12^ vg/mL) compared to Luxturna (5 × 10^11^ vg/mL) and a possible saturation effect of AAV vector loss within the cannulas. Alternatively, a difference in retention rate between serotypes can also be related to a different capsid surface structure that dictates binding properties [17,18,19].

The comparison of the four different injection cannulas revealed minor variances in their retention rates, ranging from 2.6% to 32.3%. For VN, the Disposable Dual bore BSS injection needle, D.O.R.C. (1270.06), and for the AAV8-based IND, the Disposable subretinal injection cannula, D.O.R.C. (1270.04), were associated with the lowest retention rate. However, none of the injection devices consistently outperformed the others when comparing the results with both serotypes, and the range of measured differences was close to the range of test–retest variability (mean 4.5%, range 2.5–20.2%). Considering these findings alongside other uncertainties in surgical delivery, such as actual bleb volume versus injected volume and the amount of reflux, which can also impact the vector delivered to the target cells, we deem the observed differences in retention rates clinically insignificant and therefore comparable [20].

This study is subject to several limitations. Firstly, variability in the foot pedal-controlled passage of the vector solution through the injection devices, as well as the test–retest variability of the assay, coupled with the relatively low number of experiments conducted, may have introduced some uncertainty. Additionally, the comparison of the serotype-dependent vector retention rate was performed within the combined dataset of different cannulas tested, which potentially adds to the transferability, but with the drawback of more variability in the comparison. Furthermore, the absence of a functional assay, such as a transduction assay, limits our ability to confirm whether the analyzed genome copy number correlates with functionally active vectors. Lastly, the study’s scope was restricted by the limited number of cannulas and vectors tested, potentially affecting the generalizability of the findings. Future studies should seek to validate these results through repeated measurements with additional serotypes in a broader range of subretinal injection devices from various manufacturers, to provide further insights into the retention rates.

In conclusion, this study shows comparable retention rates for the FDA-approved PolyTip cannula, MedOne Surgical, USA, and the three 41-gauge cannulas from D.O.R.C. International, The Netherlands, which therefore seem equally compatible for the subretinal application of the AAV2-based VN and any other AAV2- or AAV8-based gene therapy vectors.

## 4. Methods and Materials

### 4.1. AAV Vectors

Voretigene neparvovec was used with the approval of the manufacturer, to test the retention of an AAV serotype 2 vector within different injection cannulas. Left-over substance was used in the appropriate dilution for clinical application (5 × 10^11^ vg/mL). To test the retention of AAV serotype 8, a clinical-grade investigational new drug (IND) based on AAV serotype 8 currently in clinical testing was prepared as for clinical application (1 × 10^12^ vg/mL).

### 4.2. Application Devices

Four different injection cannulas from two different suppliers were used in the experiment. Details are shown in Table 1. All devices were subretinal 41-gauge cannulas with a tip made out of polyamide. In the product description of the cannulas Ref. 3219 and Ref. 1270.04, the cannulas are labeled as 38-gauge, which refers to the outer diameter. The inner diameter, however, is 41-gauge and the same as in the other cannulas. After diluting the vector to the suitable concentration for clinical use, the vector was transferred from one tube to another, simulating a subretinal injection in vitro. To maintain consistency, the injection process through the instruments was conducted using a foot pedal-controlled injection system, as employed in the clinical scenario [12].

### 4.3. Quantification of AAV Retention

VN, like all AAV-based gene therapies, contains essential building blocks that allow for the production and functioning of the vector. Next to the capsid proteins that are different for each serotype, these essential building blocks include inverted terminal repeats (ITRs) from the original AAV2 genome [21]. All other building blocks that define each specific investigational new drug (IND) can vary and would require a customized approach to quantification. As the precise nature of the specific IND contents (i.e., the DNA sequence of the expression cassette) can be commercially sensitive, quantification of the vector system can be difficult if the target sequence remains unknown. The ITR sequence, as an essential component of all AAV-based INDs, therefore lends itself to a global AAV quantification assay. Such an assay was used to quantify the AAV-based IND before and after passing through the instruments to quantify a retention rate.

A quantitative polymerase chain reaction (qPCR) was performed, following the design and validation of respective primer pairs used to amplify the target sequence within the vector genome. A DNA polymerase and intercalating DNA dye (SYBR Green I Master) were used together with a primer, which targeted the AAV ITR region as described by Aurnhammer et al. [22]. The samples were inactivated with 0.5625% SDS and heated for 15 min at 65 °C. The standard curve was created with an rAAV transfer plasmid, pAAV-CMV-eGFP-WPRE (internal ID P2352), containing two AAV2-ITRs. It was digested with PvuII, which cuts at both ends of each ITR, allowing better access for the DNA polymerase, allowing more precise and accurate measurements [23]. The measurement was performed in 2 dilutions, at 4.0 × 10^3^ and 4.0 × 10^4^ dilution factors. It was performed once with technical duplicates and repeated one time with technical triplicates. The titers, measured at vector genomes per mL (vg/mL), were calculated based on the copies/reaction in the standard curve and the subsequent Cp values. The standard curve covered a range from 5.0 × 10^2^ to 5.0 × 10^8^ copies/reaction.

### 4.4. Statistical Analysis

For the statistical analysis, Prism 10 for Macintosh, Version 10.1.1(270), 21 November 2023 was used. Descriptive statistics were used to summarize the data. An unpaired two-sided T test was performed to compare the retention rates between the serotypes. Continuous data is given as the mean ± SD. A *p* value of < 0.05 was considered statistically significant.

## Figures and Tables

**Figure 1 ijms-25-03705-f001:**
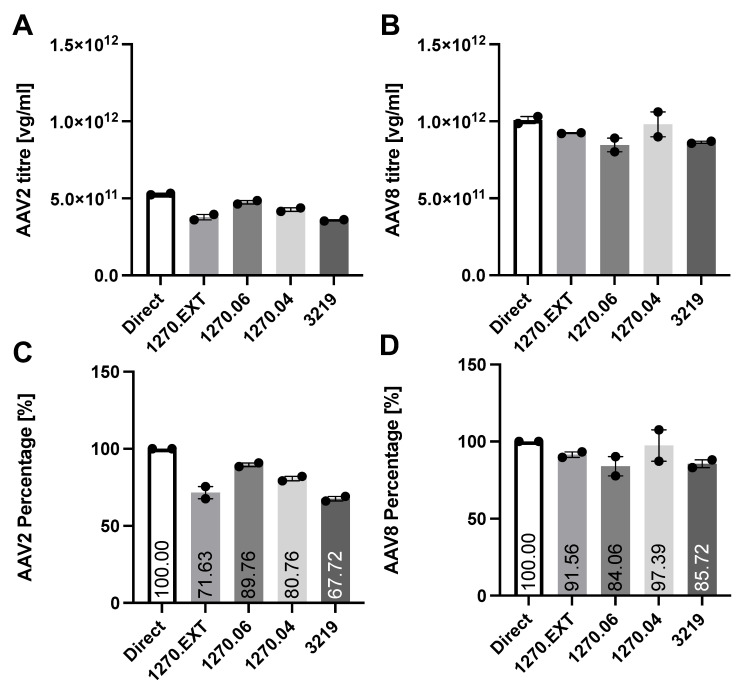
Retention of AAV Serotype 2 and of AAV Serotype 8 within the D.O.R.C. 1270.EXT, 1270.06, 1270.04, and the MedOne 3219 cannula. Absolute titers of two repeated experiments given as the average of each sample measured in two concentrations and technical triplicates on top (**A**,**B**), percentages of the direct measurement shown below (**C**,**D**).

**Figure 2 ijms-25-03705-f002:**
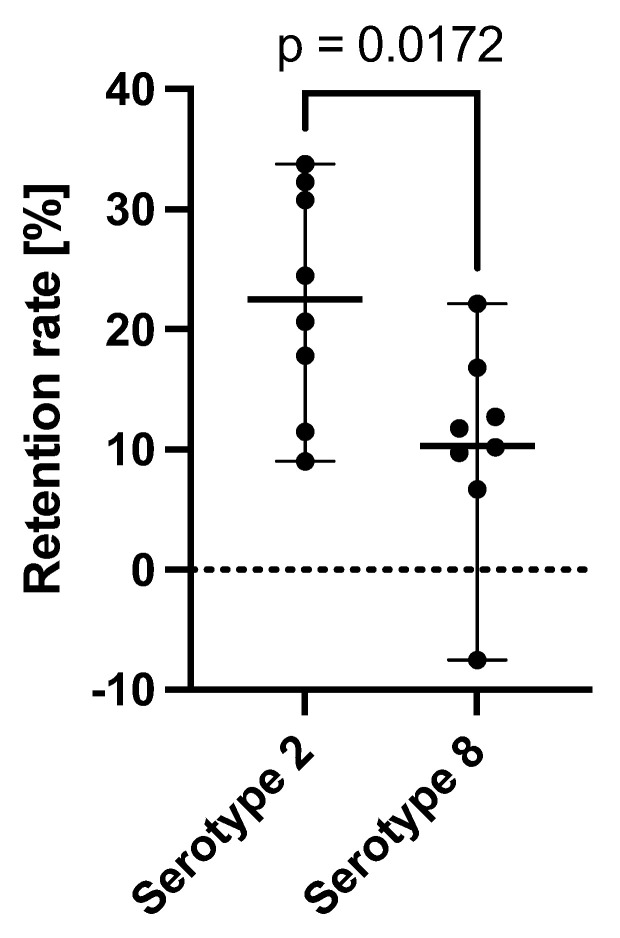
Difference in retention rate measured as [100 − (direct concentration/cannula concentration)] for AAV2-based voretigene neparvovec (Luxturna) and an AAV8-based investigational new drug within all different injection systems.

**Figure 3 ijms-25-03705-f003:**
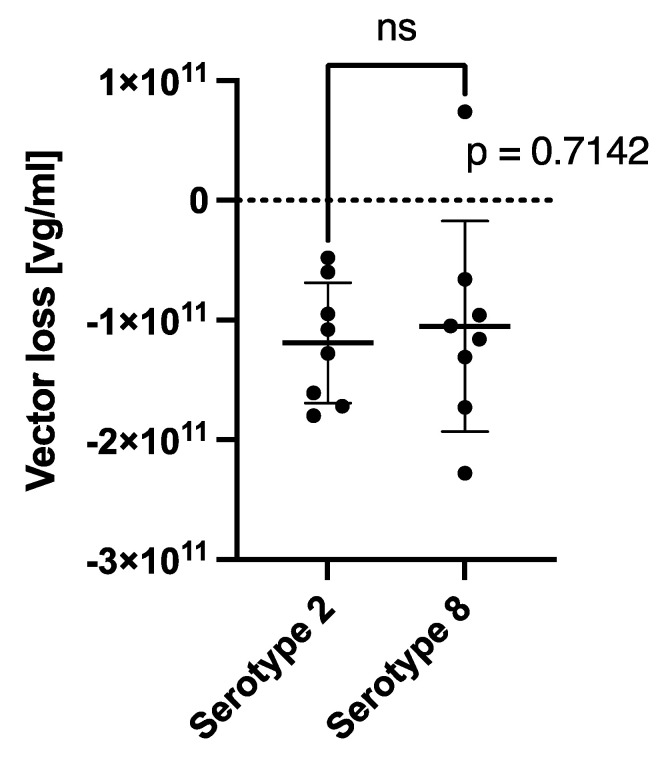
Total vector loss measured as [direct concentration − cannula concentration] for AAV2-based voretigene neparvovec (Luxturna) and an AAV8-based investigational new drug within all different injection systems.

**Table 1 ijms-25-03705-t001:** Injections cannulas tested for AAV retention rates.

Product Description	Trocar Cannula System [Minimum Gauge Size]	Manufacturer	Reference Number
PolyTip^®^ cannula 25 g/38 g	25	MedOne Surgical, Inc. Sarasota, FL, USA	3219
Extendible 41G subretinal injection needle. (23-gauge/0.6 mm)	23	D.O.R.C. International, 3214 ZG Zuidland, The Netherlands	1270.EXT
Disposable Dual bore BSS injection needle. (41-gauge/0.1 mm)	23	D.O.R.C. International, 3214 ZG Zuidland, The Netherlands	1270.06
27G Disposable subretinal injection cannula (38G)	27	D.O.R.C. International, 3214 ZG Zuidland, The Netherlands	1270.04

## Data Availability

The authors confirm that the data supporting the findings of this study are available within the article.

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
