# Peer review of "Retention Rates of Genetic Therapies Based on AAV Serotypes 2 and 8 Using Different Drug-Delivery Materials"

_ijms, 2024, doi:10.3390/ijms25073705_

Round 1
Reviewer 1 Report
Comments and Suggestions for Authors
The “Luxturna” gene therapy targeting RPE65 (VN) was the first gene therapy treatment to be approved by FDA. It is delivered via an AAV vector. Because only 2 cannulas have been listed in the manual for VN delivery, the authors compared the retention rates of two different viruses (AAV2 and AAV8), using the Poly/Tip cannula 25g/38g (MedOne surgical), and 3 different subretinal injection needles produced by DORC: that have a 41G cannula tip. The vector solution was injected through the cannulas using a foot-pedal-controlled injection system. Virus genome was quantified using qPCR for the AAV based IND before and after passing through the instruments to quantify the retention rate. For AAV2, 5 x 10e11 was used as the standard, for AAV8 1 x 10e12 was used as standard. The measurement was performed with 2 dilutions, once with technical duplicates and repeated one time with technical triplicates.
The authors concluded that all injection cannulas showed comparable retention rates.
General comments:
The is a well-written short manuscript. However, the authors should explain their methods better for the reader to understand it. The number of experiments performed are on the low side; and there should have been more done to make this convincing. Apparently, this was an in vitro simulation of a subretinal injection (from one tube to the other) – or did the reviewer miss something? The authors admit the limitations of their study in the discussion.
Specific comments:
Results:
Figure 1: it would be helpful to also show the individual retention rates as panels E) and F) underneath C) and D). The retention rates listed in the text (p. 2, bottom) do not match what one can get from figure 1 when calculating 100 minus the corresponding values in the graph.
Figure 2 and 3: because there were relatively few experiments, the authors are lumping results from all different cannulas together.

Reviewer 2 Report
Comments and Suggestions for Authors
The manuscript entitled ‘Retention rates of genetic therapies based on AAV serotype 2 and 8 using different drug-delivery materials’ requires minor revision before possible publication.
1. What do authors mean by ‘none-integrating’?
2. Line 190: Check the full form of qPCR.
3. Why is it written as 2 double stranded?
4. Line number 140/ Discussion: Here, authors have compared the retention rate and provided a range of 0.8-32.6%, but in the later part, authors concluded no one is better. It's a bit confusing because the range is very high.
5. The authors are suggested to briefly explain the future perspective of this study or how this observation can be extended to the next level.
Comments on the Quality of English LanguageIt can be improved.
